# The Effect of High Pressure Homogenization on the Structure of Dual-Protein and Its Emulsion Functional Properties

**DOI:** 10.3390/foods12183358

**Published:** 2023-09-07

**Authors:** Meishan Wu, Xiaoye He, Duo Feng, Hu Li, Di Han, Qingye Li, Boya Zhao, Na Li, Tianxin Liu, Jing Wang

**Affiliations:** 1Institute of Food and Nutrition Development, Ministry of Agriculture and Rural Affairs, Beijing 100081, China; 2The Key Laboratory of Food Resources Monitoring and Nutrition Evaluation, Ministry of Agriculture and Rural Affairs, Beijing 100081, China

**Keywords:** dual-protein, high-pressure homogenization, protein structure, emulsion functional properties

## Abstract

It has been proven that high-pressure homogenization (HPH) could improve the functional properties of proteins by modifying their structure. This study researched the effect of HPH on the structural and functional properties of whey-soy dual-protein (Soy Protein Isolation-Whey Protein Isolation, SPI-WPI). Different protein solution samples were treated with HPH at 30, 60, 90, 120 and 150 MPa, and the structure changed under different pressures was analyzed by measuring particle size, zeta potential, Fourier infrared spectrum (FTIR), fluorescence spectrum and scanning electron microscope (SEM). The results showed that HPH significantly reduced the particle size of SPI-WPI, changed the secondary and tertiary structures and improved the hydrophobic interaction between molecules. In addition, HPH significantly improved the solubility and emulsification of all proteins, and the improvement effect on SPI-WPI was significantly better than SPI and WPI. It was found that SPI-WPI treated with 60 MPa had the best physicochemical properties. Secondly, we researched the effect of HPH by 60 MPa on the emulsion properties of SPI-WPI. In this study, the SPI-WPI had the lowest surface tension compared to a single protein after HPH treatment. The emulsion droplet size was obviously decreased, and the elastic properties and physical stability of SPI-WPI emulsion were significantly enhanced. In conclusion, this study will provide a theoretical basis for the application of HPH in modifying the structure of dual-protein to improve its development and utilization in liquid specialty food.

## 1. Introduction

Soy protein isolation (SPI) is recognized as the only full-priced premium plant protein similar to animal protein, which has high nutritional value. Generally, SPI can be divided into 2S, 7S, 11S and 15S protein components, and mainly composed of 7S and 11S protein components that account for more than 70% of the total components [1,2]. Moreover, SPI is also an important raw material and additive in the food industry. It has excellent processing functional properties such as gelation, water retention, emulsification and foaming, and is widely used in meat products, baked goods, and dairy processing [3,4].

Whey protein isolation (WPI) is a kind of small molecular globular protein isolated from milk during cheese processing and is mainly composed of β-Lactoglobulin (β-Lg), α-Lactalbumin (α-La), Lactoferrin (LF), bovine serum albumin (BSA) and Immunoglobulin (Ig). β-Lg and α-La occupy the highest proportion of WPI, accounting for 50% and 25% of the total components, respectively [5,6]. WPI is a high-quality animal protein with good digestion and absorption. In addition, WPI is also used in food processing, where its good gelation, emulsification and foaming make it widely applied to meat and dairy product processing [7].

In recent years, animal-plant mixed proteins have gradually attracted the attention of many researchers. The whey-soy ‘dual-protein’, discovered by studying the dose-effect and mechanisms of synergism and interactions of different animal-plant mixed proteins, has a complete range of amino acids and contains rich essential amino acids. Its high nutritional value and bioavailability play an important role in maintaining body health and promoting body growth. Most importantly, the preliminary findings demonstrated that the dual-protein had significant nutritional quality improvement in hematopoietic reconstruction, immune function, muscle level, lipid metabolism, and osteoporosis [8,9,10].

High-pressure homogenization (HPH) is a kind of non-thermal processing technique that has been widely used in food processing. It has the advantages of low energy consumption, low cost, simple operation, high efficiency, and can avoid the problems of sensory quality destruction and nutrient loss caused by thermal processing [11]. Originally, HPH was only used to homogenize complex liquid products such as fruit and vegetable juices to enhance their physical stability as a standardization step [12,13]. The working general principle of HPH can be summarized as follows: the liquid material is transported to the closed homogeneous valve area by the HPH machine plunger pump at high pressure and low flow rate. Under the action of high pressure, the liquid material quickly flows through the adjustable flow limiting the homogeneous valve gap. At this time, it is subjected to mechanical forces such as high-speed shear, shock, and cavitation phenomenon, resulting in a series of comprehensive effects such as pressure loss, expansion, shear, and high-speed impact, which makes the liquid particles crushed and destroyed microbial cells [14]. HPH has a side effect of temperature increase in the valve and product, dependent on the height of pressure drop within the valve. Generally, the product temperature rises by 2.0–3.2 °C for every 10 MPa of pressure using HPH; therefore, it is necessary to use a cooling system to maintain the product’s original quality [15].

The mechanical and cavitation effects produced by HPH can significantly reduce the particle size of protein molecules, improve the electrostatic interaction between protein molecules, modify the protein structure, and effectively improve the physicochemical properties [16]. Zhao et al. [17] found that HPH could change the structure of SPI from spherical to loose lamella, and the antioxidant activity of SPI hydrolysate was significantly increased. Melchior et al. [18] found that the content of β-sheet in the secondary structure of pea protein decreased significantly after HPH treatment, which improved the digestibility of pea protein in vitro. Ma et al. [19] also found that HPH improved the surface hydrophobicity of cod protein, improved its emulsifying, and promoted the formation of the elastic modulus of the emulsion interface.

Protein modification by HPH has gradually become a research hot spot, while the structure and functionality of dual-protein upon HPH treatment have not been extensively studied so far. In this study, the structure of dual-protein was investigated, and the emulsion functional properties were assessed.

## 2. Materials and Methods

### 2.1. Materials

SPI (protein content, 90%) was obtained from IFF Inc. (Shanghai, China). WPI (protein content, 95%) was obtained from Agropur Cooperative (Montreal, QC, Canada). Soy oil was purchased from Yihai Karry Co., Ltd. (Shanghai, China). All reagents were of analytical-grade purity.

### 2.2. Preparation of Protein Solution and HPH Treatment

The protein powder (SPI and WPI) were dispersed in de-ionized water at a total protein concentration of 6% (*w/v*) to achieve SPI, WPI, dual-protein (SPI-WPI, 1:1) solutions, and they were stirred at room temperature for 1.5 h. After that, all the solutions were stored at 4 °C for 12 h to ensure full hydration. After the protein solutions were restored to room temperature (25 °C), they were filtered by a 0.3 mm sieve and were homogenized for three cycles at 30, 60, 90, 120 and 150 MPa by using a high-pressure homogenizer (Panda PLUS 2000, GEA Group, Düsseldorf, Germany), each cycle time was approximately 13 s. The high-pressure homogenizer was connected with a condenser, to keep the temperature of protein solutions at 18 ± 1 °C after each HPH treatment cycle. Untreated protein solutions were used as a control.

### 2.3. Dual-Protein Structure Measurement

#### 2.3.1. Particle Size and Zeta Potential Measurement

All the protein solutions were diluted to 0.6 mg/mL by using de-ionized water, and the particle size and zeta potential were measured by using a nanoparticle size analyzer (Zetasizer Nano-ZS90, Marvin Instrument Co., Ltd., Marvin, UK) at room temperature according to the previous report [20].

#### 2.3.2. Solubility Measurement

The solubility of protein solutions was measured by the Bradford method, and BSA was used as the standard. All the protein solutions were centrifuged at 10,000 rpm for 30 min and then diluted the supernatant to 0.06 mg/mL by using de-ionized water. The diluted supernatant was mixed with Coomassie brilliant blue G-250 at a volume ratio of 1:5 for 2 min, and the absorbance was measured by using an ultraviolet-visible spectrophotometer (UV1780, Shimadzu Corporation, Kyoto, Japan) at 595 nm. By substituting the absorbance of the supernatant into the BSA standard curve, the protein concentration of the supernatant was calculated. Finally, the solubility was calculated based on the ratio of the protein concentration in the supernatant to the original protein concentration before centrifugation.

#### 2.3.3. Sodium Dodecyl Sulfate-Polyacrylamide Gel Electrophoresis (SDS-PAGE)

According to the method of Laemmli et al. [21], the molecular weight distribution of all the samples was measured by SDS-PAGE; 16% separation gel and 4% concentration gel were used. The samples were centrifuged at 15,000 rpm for 30 min, then mixed with loading buffer at 1:3. The mixed solution was heated in a water bath at 90 °C for 5 min. The conditions of electrophoresis were set at 125 V for 80 min by using gel electrophoresis apparatus (DYY-6C, Bio-Rad Laboratories, Hercules, CA, USA). When the electrophoresis was finished, the film was stained with Coomassie brilliant blue R-250 for 60 min, then decolorized with Isopropyl alcohol-glacial acetic acid until transparent. The molecular weight distribution result was observed by protein marker after SDS-PAGE.

#### 2.3.4. Fourier Transform Infrared Spectroscopy (FTIR)

The protein samples were freeze-dried; 1 mg freeze-dried samples was mixed with 100 mg KBr, and they were pressed to a slim film before measurement. We used an FTIR Spectrometer (VERTEX 70V, Bruker Corporation, Karlsruhe, Germany) to carry out full-band scanning with a scanning range of 4000–400 cm^−1^, a scanning rate of 4 cm^−1^, and a scanning time of 16. The secondary structure was fitted and calculated by using OMNIC 8.2 software.

#### 2.3.5. Intrinsic Fluorescence Spectroscopy

The intrinsic fluorescence spectrum of the protein samples was measured by the Fluorescence Spectrophotometer (F-7000, Hitachi Ltd., Tokyo, Japan). According to Han et al. [22], the emission wavelength was set at 300–450 nm, the excitation wavelength at 290 nm, and slit width at 5 nm.

#### 2.3.6. Surface Hydrophobicity

According to Kato et al. and Cui et al. [23,24], we used the 8-anilinonaphthalene-1-sulfonate (ANS) as a fluorescent probe to measure the surface hydrophobicity of protein samples. The protein solutions were diluted to 0.2, 0.4, 0.6, and 0.8 mg/mL by using 10 mmol/L phosphate buffer solution (PBS, pH 7.0). Subsequently, 30 μL of 8 mmol/L ANS solution and 3 mL diluted samples were mixed for 5 min in the dark. The surface hydrophobicity index (H_0_) was measured by using the fluorescence spectrophotometer (F-7000, Hitachi Ltd., Tokyo, Japan), with an excitation wavelength of 380 nm, emission length of 470 nm, and slit width of 5 nm. The protein concentration was the horizontal axis, the fluorescence intensity was the vertical axis, and the slope of the curve was the H_0_ of protein samples.

#### 2.3.7. Microstructure

The microstructure of protein samples was observed by using a scanning electron microscope (SEM, S-4800, Hitachi Ltd., Tokyo, Japan). Freeze-dried protein samples were treated with gold spray, and then put on the SEM stage. The microstructure of samples was observed under the 5 kV scanning voltage. The type of detection used was secondary electrons.

### 2.4. Dual-Protein Emulsion Properties Measurement

#### 2.4.1. Emulsifying Activity Index (EAI) and Emulsifying Stability Index (ESI)

The EAI and ESI were measured according to the previous method reported by Pearce et al. [25]. 25% (*v*/*v*) soy oil was added to the protein solution, and mixed at 10,000 rpm for 2 min by using a high-speed homogenizer (Ultra Turrax T25 Digital, IKA-Werke GmbH & Co. KG, Stuttgart, Baden-Württemberg, Germany); 80 μL emulsion was added to 8 mL 0.1% (*w*/*v*) SDS solution immediately, mixed for 2 min. The absorbance was measured by using an ultraviolet-visible spectrophotometer (UV1780, Shimadzu Corporation, Kyoto, Japan) at 500 nm. After 10 min, we repeated the same operation.

EAI and ESI were calculated as:(1)$$\mathrm{EAI}\left(\frac{m^{2}}{g}\right)=2\times 2.303\times \frac{A_{0}}{c}\times \left(1-\varphi \right)\times 100\%$$
(2)$$\mathrm{ESI}\left(\%\right)=\frac{A_{0}}{A_{10}}\times 100\%$$

In Equations (1) and (2), A_0_ and A_10_ were the absorbance values at 500 nm for 0 min and 10 min, respectively. φ was the volume fraction of the soy oil (25% *v*/*v*). c was the concentration of protein (g/mL).

#### 2.4.2. Surface Tension

The SPI, WPI and SPI-WPI solutions [6% (*w/v*)] were homogenized for three cycles at 60 MPa by using a high-pressure homogenizer (Panda PLUS 2000, GEA Group, Düsseldorf, Germany). The surface tension of the protein samples was measured by using the surface tension meter (Surface Tension Analyzer, Biolin Scientific, Gothenburg, Sweden). Then, the protein solutions were added to the glass at 45 mL as the water phase, the platinum plate was dipped into the solutions and the soy oil was added to the top of the solutions as the oil phase. Finally, the dynamic surface tension was characterized by recording the platinum plate rising from the water phase to the water-oil phase junction, and the recording time was kept for 3600 s.

#### 2.4.3. Emulsion Preparation

We mixed soy oil with protein solutions at 1:9 (*w*/*w*) by using a high-speed homogenizer (Ultra Turrax T25 Digital, IKA-Werke GmbH & Co. KG, Stuttgart, Baden-Württemberg, Germany) at 5000 rpm for 1 min. Then all the O/W protein emulsions were processed by three cycles of HPH treatment at 50 MPa by using a high-pressure homogenizer (Panda PLUS 2000, GEA Group, Düsseldorf, Germany).

#### 2.4.4. Droplet Size and Zeta Potential of Emulsion

The droplet size and zeta potential were measured by using a nanoparticle size analyzer (Zetasizer Nano-ZS90, Marvin Instrument Co., Ltd., Marvin, UK). All the emulsions were diluted to 0.12 mg/mL with de-ionized water and measured at room temperature.

#### 2.4.5. Physical Stability of Emulsion

Using a Tubiscan stability analyzer (Tubiscan Tower, Formulaction, Toulouse, France) to measure the physical stability of protein emulsions. According to the previous method reported by Essam et al. [26], the back-scattered light was set to measure at 15 min intervals for 5.75 h, and all the tests were measured at room temperature. The changes in backscattering profiles and Turbiscan stability index (TSI) were calculated by using Towersoft 2.0.0.9.

#### 2.4.6. Micro-Rheological of Emulsion

The micro-rheology of protein emulsions was performed by using an optical micro-rheometer (Rheolaser Master, Formulaction, Toulouse, France). According to the previous method described by Zhu et al. [27], 20 mL protein emulsion was added into a glass tube, and then placed in the Rheolaser chamber. Measuring at 5 s intervals for 20 min and all the tests were measured at room temperature. The elasticity index (EI), macro viscosity index (MVI) and solid-liquid balance (SLB) of protein emulsions were obtained by using Rheosoft Master 1.4.0.0.

### 2.5. Statistical Analysis

All the data statistical differences were analyzed by using SPSS 26 (IBM Corporation, Armonk, NY, USA) through a one-way analysis of variance (ANOVA) test, and Origin 2021 was used for data analysis and visualization. The differences were considered significant at *p* < 0.05.

## 3. Results and Discussion

### 3.1. Particle Size and Zeta Potential

The changes in appearance of the protein solutions after HPH treatment are shown in Figure 1a. The sediment occurred in SPI and SPI-WPI samples after standing for 24 h, while there was no sediment in SPI and SPI-WPI samples after HPH treatment, indicating that HPH improved the uniformity of SPI and SPI-WPI solutions. Most importantly, all the WPI solutions were clear and transparent in appearance and did not change significantly before and after placement.

The particle size and zeta potential have significant effects on the stability and physicochemical properties of the protein solution system. The sediment in the untreated SPI and SPI-WPI samples might cause measurement error; Figure 1b,c showed the protein particle size and zeta potential results after HPH treatment separately. The particle size of SPI-WPI was smaller than SPI but obviously larger than WPI, and the zeta potential also had a similar finding. It could be seen that the particle size of all protein solutions decreased gradually with the increase in pressure. Previous studies have reported that the high shear force caused by cavitation and turbulence during HPH treatment could destroy the structure of protein molecules, causing the decomposition of large molecules into smaller particles [28]. In particular, the smaller particle size could promote the more stable protein solutions, thus indicating that HPH might have a positive effect on the stability of protein solutions.

The greater the absolute value of zeta potential, the stronger the electrostatic interaction between protein molecules [29]. All the samples showed a negative zeta potential value. The absolute value of the zeta potential of SPI and WPI both showed the biggest value at 90 MPa and decreased as the pressure increased further, suggesting that the decline in particle size might lead to more net surface charge of the molecules. However, the denaturation of protein molecules under excessive pressures also resulted in an electrostatic repulsion decrease. In addition, the zeta potential of SPI-WPI had no significant difference, suggesting that HPH had no significant effect on the zeta potential of SPI-WPI.

### 3.2. Solubility

Figure 1d shows the solubility of protein under different homogenization pressures treatment. As the pressure increased from 0 to 60 MPa, the solubility of SPI significantly increased from 27.17% to 49.17%. The solubility of untreated SPI-WPI was significantly higher than untreated SPI, and its solubility also increased to 74.84% as the pressure was 60 MPa. This phenomenon was attributed to HPH destroying hydrogen bonds and hydrophobic interaction, reducing the particle size. Therefore, the smaller particle size could provide a bigger surface area to promote the interaction between protein molecules and water [30]. In contrast, the solubility of SPI and SPI-WPI decreased as the pressure exceeded 60 MPa, which might be related to insoluble re-aggregation between protein molecules caused by hydrophobic interaction as the pressure increased further [31]. Untreated WPI had the biggest solubility among the samples, while its solubility decreased from 93.09% to 74.50% as the pressure increased from 0 to 150 MPa. This tendency could be explained by the decline of electrostatic repulsion, which led to the aggregation of WPI molecules and reduced solubility [32]. It is worth noting that the solubility of SPI-WPI increased from 69.33% to 72.59% as the pressure increased from 90 to 150 MPa, which might be due to the hydrophilic structure being exposed under the higher pressure treatment.

### 3.3. SDS-PAGE

Figure 1e shows the protein molecule weight distribution under different homogenization pressures through SDS-PAGE analysis. As we can see, untreated SPI just showed acidic polypeptide chains A of soy 11S globulin, after HPH treatment, all the SPI samples showed α, α′, β subunits of soy 7S globulin and A, B polypeptide chains of soy 11S globulin, which indicated that the solubility of SPI increased under HPH treatment. As the pressure increased, the color of basic polypeptide chains B of soy 11S globulin turned to light, while the β subunits of soy 7S globulin and acidic polypeptide chains A of soy 11S globulin turned to deep. Furthermore, there were some small bands that can be seen from 60 to 70 KDa when the homogenization pressure increased to 150 MPa, reflecting that HPH treatment promoted the formation of high molecule weight aggregates of SPI. As for WPI, it could be observed in molecule weight bands of β-Lactoglobulin (β-Lg) and α-Lactalbumin (α-La), and there was no obvious difference among different homogenization pressures, indicating that HPH barely affected the structure of WPI. The molecule bands of untreated SPI-WPI not only included WPI, but also α subunits of soy 7S globulin and acidic polypeptide chains A of soy 11S globulin, besides a few bands above 55 KDa appeared. It could be seen that all major molecule bands of SPI and WPI appeared in SPI-WPI under HPH treatments, and there was no obvious difference among different homogenization pressures. Different from the single SPI, the bands near the subunits of soy 7S globulin became more intensive, indicating that HPH promoted the protein interaction of SPI-WPI, and formed the aggregation structure with bigger molecule weight.

### 3.4. FTIR

The amide I band (1600–1700 cm^−1^) can represent the C=O and C-N stretching vibration, which is used to analyze the secondary structure of proteins [33]. Figure 2 shows the relative content of the secondary structure of proteins through curve fitting of the amide I band. The majority of secondary structures of SPI were β-sheet and β-turn, and the relative content of α-helix increased from 14.91% to 17.86% under 30 MPa treatment, indicating that the hydrogen bond interaction enhanced, which informed the highly ordered protein structure [34]. As the pressure increased to 90 MPa, the relative contents of α-helix and random coil decreased from 17.86% to 14.48% and from 14.19% to 12.94%, while the β-sheet and β-turn increased from 37.60% to 38.24% and from 32.76% to 34.34%, suggesting that the α-helix and random coil were transformed to β-sheet and β-turn, the structure of SPI was changed. When the pressure increased to 150 MPa, the relative contents of the random coil obviously increased, and the loss of α-helix and β-turn reflected that the molecule was unfolded and enhanced the protein flexibility. The α-helix of WPI obviously increased under HPH treatment, but the β-sheet, β-turn, and random coil decreased at different intent. The α-helix was the most stable structure; therefore, the secondary structure of WPI was transformed from a loose and disordered structure to a more compact and stable spiral structure. Moreover, the relative contents of α-helix of untreated SPI-WPI were more than single SPI and WPI, it could be seen that the hydrogen bond interaction improved. The relative contents of the α-helix increased further as the pressure increased to 60 MPa while decreasing to 17.26% at 150 MPa. The change in the random coil was opposite to the α-helix, the structure of SPI-WPI changed and the spatial structure became more stretched. In addition, the relative contents of the β-turn also increased from 29.99% to 32.79% as the pressure increased to 90 MPa and decreased obviously from 32.79% to 29.31%, it also could be explained by the hydrogen bond interaction becoming weakened as the pressure increased further.

### 3.5. Intrinsic Fluorescence Analysis

Tryptophan, tyrosine and other chromophoric groups exist in the side chain of proteins, and intrinsic fluorescence can be generated by applying certain excitation wavelengths, so it is often used to characterize the tertiary structure changes of proteins [35]. Figure 3a–c shows the intrinsic fluorescence change of SPI, WPI and SPI-WPI under different homogenization pressure treatments. The max fluorescence intensity of untreated SPI was 2439, which increased significantly to 3360 as the pressure increased to 90 MPa, indicating that more chromophore groups were exposed. As the pressure increased further, the max fluorescence intensity decreased gradually to 3357. In addition, the max fluorescence wavelength of SPI also blue-shifted from 334 nm to 331 nm as the pressure increased to 90 MPa, and red-shifted to 332 nm under 150 MPa treatment. HPH enhanced the fluorescence intensity of SPI and changed the polar environment of the chromophores groups. The max fluorescence intensity of WPI obviously decreased from 2912 to 2503 after HPH treatment, and the max fluorescence wavelength had no significant change, which was similar to what Shi et al. [36] described before. The fluorescence intensity of SPI-WPI significantly punched from 1763 to 2872 as the pressure increased to 60 MPa, then slightly decreased to 2788 under 120 MPa and last obviously increased to 2889 under 150 MPa, the wavelengths also blue shift from 335 nm to 333 nm. Compared to SPI and WPI, the rising trend of fluorescence intensity of SPI-WPI was significantly higher than single protein, suggesting that HPH had more effects on the tertiary structure of SPI-WPI and increased the exposure of chromophores groups after treatment.

### 3.6. Surface Hydrophobicity Analysis

ANS is a common fluorescent probe, which can be non-covalently bonded with hydrophobic groups in protein molecules in an aqueous solution, resulting in a significant increase in fluorescence intensity and allowing further characterization of the change in protein surface hydrophobicity [37]. The H_0_ could be calculated by the slope of each curve of protein which could be found in Appendix A. Figure 3d shows the H_0_ of SPI, WPI and SPI-WPI under different homogenization pressure treatments. The H_0_ of SPI grew from 1866.75 to 2355.25 as the pressure increased to 90 MPa, which could explain the hydrogen bond interaction being disrupted; contributing to the exposure of internal hydrophobic groups [38]. When the pressure increased to 150 MPa, the reduction of H_0_ might be related to the aggregation of the molecules formed and the decreased surface hydrophobicity. However, the H_0_ of WPI decreased from 1004.50 to 915.25; the lessening of H_0_ may be relative to the hydrophobic region on the surface reduced due to aggregation of WPI, which resulted in the reduction of binding to the fluorescent probe. On the contrary, the H_0_ of SPI-WPI climbed with an obvious tendency from 1532.60 to 1915.95 as the pressure increased from 0 to 150 MPa. This phenomenon showed that HPH treatment was easier to expand the molecule structure of SPI-WPI compared to single SPI and WPI. We found that the higher the homogenization pressure of the treatment, the more significant the degree of molecule space structure opening and the promotion of the hydrophobic interactions.

### 3.7. Microstructure

Figure 4 shows the microstructure change of SPI, WPI and SPI-WPI under different homogenization pressures. After freeze-drying, all protein samples showed lamellar structure. It was because HPH destroyed the dense spherical structure of the protein and opened the peptide chain, which formed an irregular agglomeration after freeze-drying, and the texture became loose. The untreated sample had a rough structure, after HPH treatment, the pore structure of the sample became more obvious, and the pore distribution tended to be more uniform and dense with the increase in pressure. Compared with a single protein, the pore structure in SPI-WPI was denser under the same HPH condition, these results indicated that HPH made the interaction and crosslinking degree among SPI-WPI more significant than a single protein.

### 3.8. Emulsion Properties

#### 3.8.1. Emulsifying Activity Index (EAI) and Emulsifying Stability Index (ESI)

Proteins can spontaneously diffuse to the oil-water interface in an oil-water system, where hydrophobic and hydrophilic groups are bound to oil droplets and water molecules, respectively, to form a firm adsorption layer [39]. Figure 5a,b showed the EAI and ESI of SPI, WPI and SPI-WPI under different homogenization pressures that were calculated by Equations (1) and (2), respectively. HPH obviously improved the EAI of SPI, WPI and SPI-WPI, which increased to the maximum at 60 MPa simultaneously. These results could explain that the cavitation shear effect of HPH reduced the particle size of protein and promoted the interaction with water. At the same time, the exposure of hydrophobic groups also improved the hydrophobic interaction, which can be more firmly adsorbed on the oil-water interface [29]. On the other side, the EAI decreased at higher pressure because the reduction of hydrophobic interaction and solubility declined. The ESI of SPI and SPI-WPI also reached a maximum of 60 MPa. However, the ESI of WPI decreased slightly after HPH treatment, which might be due to the molecule denaturation that cannot be stably adsorbed on the oil-water interface. The results of EAI and ESI indicated that the combination of SPI and WPI could improve the emulsifying of a single WPI.

#### 3.8.2. Surface Tension and Interface Adsorption Kinetics

Protein would be adsorbed to the oil-water interface in the process of emulsifying oil droplets, and the interfacial tension would gradually decrease. Figure 5c,d showed the surface tension (σ) and surface pressure (π) of SPI, WPI and SPI-WPI. In the first 250 s, the surface tension of all proteins rapidly decreased and the interfacial pressure rapidly increased. The surface tension decreased gradually and the surface pressure increased slowly with the extension of time, and the surface tension still decreased at 3600 s, indicating that all protein molecules adsorbed on the oil-water interface did not reach the equilibrium state. The initial surface tension of SPI-WPI was lower than that of a single protein, indicating that SPI-WPI had the ability to significantly reduce surface tension in the adsorption process, and could be adsorbed on the oil-water interface more rapidly. HPH significantly reduced the surface tension of SPI but decreased the surface tension of WPI and SPI-WPI. It may be related to the molecules aggregation and unaggregated protein molecules absorbed on the interface at the same time, which hindered the diffusion and movement in the water phase and affected the adsorption rate at the oil-water interface [40].

The adsorption kinetics of proteins at the oil-water interface can be roughly divided into three stages: adsorption, penetration and reorganization [41]. Proteins transfer from the water phase to the oil phase through diffusion, adsorption and penetration occur at the interface, which leads to conformational reorganization. Adsorption equilibrium is also reached through the formation of the interface network structure involving disulfide bonds, hydrogen bonds, and hydrophobic interaction [42]. Generally, using the equation modified by Ward and Tordai could calculate the protein diffusion phase [43]:(3)$$\Pi =2C_{0}K_{B}T{\left(D_{t}/3.14\right)}^{1/2}$$

In Equation (3), Π is surface pressure, C_0_ is initial protein concentration, K_B_ is Boltzmann constant, T is absolute temperature, D is diffusion coefficient, and t is absorption time. If the diffusion of the emulsifier at the interface dominates the adsorption kinetics, the Π-t^1/2^ should be linear, and the slope is denoted by K_diff_ [44]. As the surface pressure increased further, the emulsifier-emulsifier interactions caused the linear relationship of the curve to decrease, it could be investigated by the method mentioned by Graham and Phillips [45]:(4)$$\ln\left(\Pi_{f}-\Pi_{t}\right)/\left(\Pi_{f}-\Pi_{0}\right)=-k_{i}t$$

In Equation (4), Π_f_, Π_t_, Π_0_ are the final adsorption time, initial adsorption time and surface pressure at any moment, respectively, and k_i_ is the first-order rate constant. The resulting curve usually contains two or more linear regions. The slope in the first linear region corresponds to the first-order rate constant of permeability (K_p_), while the slope in the second linear region corresponds to the first-order rate constant of rearrangement (K_r_) [44].

Figure 5e shows the surface pressure as a function of the square root of time (t^1/2^) for the protein diffused from the water phase to the oil phase in the early stage of absorption. Figure 5f shows two linear regions with slopes corresponding to the rate constants of permeability and rate constants of rearrangement. All the samples showed that the Π-t^1/2^ were a linear relationship, and the slope of permeability was higher than rearrangement at the late stage_,_ indicating that the diffusion behavior was dominant at the early stage, and the rearrangement of the molecules managed interface formation in the later stage.

Table 1 listed the K_diff_, K_p_ and K_r_ values of SPI, WPI and SPI-WPI. The K_diff_ of WPI was higher than SPI and SPI-WPI because WPI had a smaller molecule weight that could be transferred and adsorbed to the oil-water interface quickly. The K_diff_ of SPI-WPI was significantly higher than SPI, indicating that the addition of WPI could improve the ability of SPI to reduce interfacial tension. HPH treatment significantly increased the K_diff_ of WPI, while the K_diff_ of SPI and SPI-WPI decreased slightly after HPH treatment. At the late stage of interfacial formation, the K_r_ of SPI-WPI was significantly higher than single SPI and WPI, reflecting that SPI-WPI was easier to expand and rearrange at the interface. After HPH treatment, the K_p_ of all protein samples increased, but the K_r_ of WPI and SPI-WPI significantly decreased. Among them, the decrease of SPI-WPI was more obvious, which may be due to the more complex structure of SPI-WPI after HPH treatment, resulting in the lower development and rearrangement speed of molecules on the interface [40].

#### 3.8.3. Emulsion Droplet Size and Zeta Potential

Figure 6a,b showed the droplet size and zeta potential of SPI, WPI and SPI-WPI emulsion containing 10% (*w/w*) oil. The mean droplet size of SPI decreased obviously from 685 nm to 547 nm, perhaps because the HPH could supply the high energy for emulsification and cut the droplet size into smaller [46]. Similar changes took place in the SPI-WPI emulsion which the droplet size decreased from 485 to 425 nm, because the high shear rate refined the emulsion droplets, and the smaller droplet size was beneficial to improve the dispersion stability of the emulsion. Whereas the mean droplet size of the WPI emulsion increased from 295 nm to 301 nm, it might be that the protein aggregation resulted in the increase in droplet size.

#### 3.8.4. Physical Stability Properties

Turbiscan stability analyzer is based on the principle of multiple heavy light scattering, using transmitted light and backscattered light sources, through the static sample after a certain period of time top-down scanning, according to the collected spectral line data to characterize the changes of the sample. It has been widely used in the study of dispersion stability of emulsion and suspension systems [47]. Figure 6c shows the backscattering change of protein emulsion containing 10% (*w*/*w*) oil by using a Turbiscan stability analyzer. Clearly, the backscattering of the SPI emulsion showed the typical sediment condition, while the WPI and SPI-WPI emulsion showed the floating condition. The backscattering of the SPI-WPI emulsion was significantly lower than the WPI emulsion, indicating that the stability of the SPI-WPI emulsion was better than the WPI emulsion. After HPH treatment, the backscattering of the SPI emulsion reduced, almost reflecting the backscattering of the left and right sides decreased, indicating that the HPH treatment improved the physical stability of the SPI emulsion. The same phenomenon could be found in the WPI emulsion, in which the backscattering of the right side decreased, and the floating tendency reduced. However, the backscattering of the right side of the SPI-WPI emulsion slightly increased after HPH treatment. It might be caused by the rate of expansion and rearrangement of complex structures at the interface decreased, the oil-water interface film cannot form in time and raised the oil floating.

The Turbiscan Stability Index (TSI) can reflect the kinetic instability of an emulsion system within a certain time. The smaller the TSI value, the better the system stability. Figure 6d shows the TSI value of protein emulsion. The TSI values of all protein emulsions increased gradually with time, SPI and SPI-WPI emulsions had the lowest TSI, which both increased to 0.2, and the TSI of WPI emulsion increased to 0.4. After HPH treatment, the TSI of SPI emulsion declined to 0.1, while the WPI and SPI-WPI emulsion increased from 0.4 to 0.6 and from 0.2 to 0.3, respectively, indicating that the HPH treatment was not good enough in improving the stability of the WPI and SPI-WPI emulsion. However, the TSI of the SPI-WPI emulsion was lower than the WPI emulsion, suggesting that the SPI-WPI emulsion had a more stable structure than the WPI emulsion after HPH treatment.

#### 3.8.5. Microrheology of Protein Emulsion

Microrheology can usually be used to trace the motion of colloidal particles at the scale of 0.1–10 μm. By detecting the Brownian motion of droplets, the displacement of particles can be accurately detected and the microrheological behavior of liquids can be analyzed [48]. The microrheology could reflect the structure and viscoelastic changes of the emulsion system by the mean square displacement (MSD) of the emulsion droplet as the formula of the de-correlation time. Figure 6e shows the microrheology of the protein emulsion of SPI, WPI and SPI-WPI. The MSD curve of SPI-WPI emulsion with the same oil phase concentration is more linear than that of single protein emulsion, indicating that the SPI-WPI emulsion had a stronger viscosity than single protein emulsion. After HPH treatment, the MSD curve distribution of SPI emulsion was further narrowed. Also, the linear relationship between the curve and the de-correlation time was reduced, and the emulsion structure changed, which was mainly reflected in the reduction of emulsion viscosity and the enhancement of elasticity. This same phenomenon could be found in the SPI-WPI emulsion after HPH treatment. However, the HPH-treated WPI emulsion showed a more significant linear relationship, which enhanced the emulsion viscosity.

Table 2 shows the elasticity index (EI), macroscopic viscosity index (MVI) and solid-liquid balance (SLB) of SPI, WPI, and SPI-WPI emulsion calculated by Rheosoft Master 1.4.0.0 software. It can be seen that the characteristics of protein emulsion were mainly elastic. After HPH treatment, the EI of SPI and WPI emulsion both decreased, while SPI-WPI emulsion increased. HPH just improved the MVI of WPI emulsion, but the MVI of SPI and SPI-WPI emulsion declined significantly, it was because the reduction in the droplets improved the emulsion fluidity, which resulted in the decrease of emulsion viscosity.

SLB is the slope value of the MSD curve plateau in microrheology, which can reflect the solid-liquid equilibrium state of the system. When SLB < 0.5, the droplet movement speed of the system is slow, which mainly shows the elastic solid behavior. When SLB = 0.5, the system is in a solid-liquid equilibrium state. When 0.5 < SLB < 1, the viscous liquid behavior is mainly manifested. When SLB > 1, sediment occurs [49]. After HPH treatment, the SLB of SPI and WPI emulsion significantly increased from 0.78 to 1.01 and decreased from 0.52 to 0.36, respectively, thus it could be seen that there was slight sediment in the SPI emulsion and solid particles in the WPI emulsion. As for the SPI-WPI emulsion, the SLB increased from 0.60 to 0.73, showing more significant viscous liquid characteristics compared to a single protein emulsion.

## 4. Conclusions

HPH significantly reduced the particle size of SPI-WPI, improved the interaction between SPI-WPI molecule subunits, and changed the secondary and tertiary structures. It mainly manifested in the transformation of α-helix to the irregular curly structure, promoted the expansion of molecular structure and enhanced the hydrophobic interaction between molecules. Microstructure observations also showed that HPH promoted SPI-WPI to form a more porous pore structure. In addition, HPH treatment also significantly improved the solubility and emulsification of SPI-WPI, and HPH treatment by 60 MPa significantly improved the physicochemical properties of SPI-WPI. Besides, the SPI-WPI both showed the lowest surface tension before and after HPH treatment, indicating that the SPI-WPI was easier to absorb the oil-water interface. The HPH treatment also reduced the emulsion droplet size of SPI-WPI and enhanced the electrostatic interaction between the droplets. The elasticity of SPI-WPI emulsion also improved after HPH treatment, and the viscosity declined simultaneously, and the emulsion tended to become liquid. This research reflected that HPH has a positive promotion to the structure and emulsion functional properties of dual-protein, which could provide scientific and technological support to the development of dual-protein liquid special medical food in the future.

## Figures and Tables

**Figure 1 foods-12-03358-f001:**
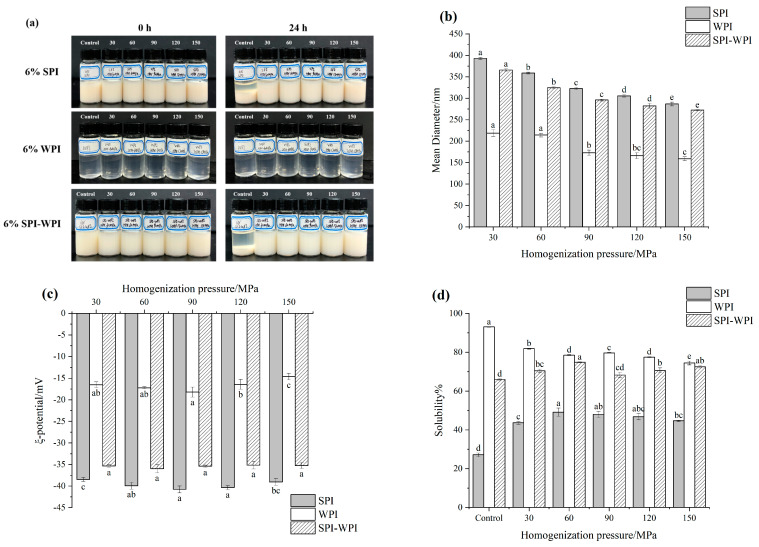
The effect of high pressure homogenization on the appearance (**a**), mean diameter (**b**), zeta potential (**c**), solubility (**d**), SDS-PAGE (**e**) of SPI, WPI and SPI-WPI. From left to right in Figure (**a**) is Control, HPH 30 MPa, HPH 60 MPa, HPH 90 MPa, HPH 120 MPa, HPH 150 MPa. Different lower case letters in Figure (**b**–**d**) indicate a significant difference (*p* < 0.05).

**Figure 2 foods-12-03358-f002:**
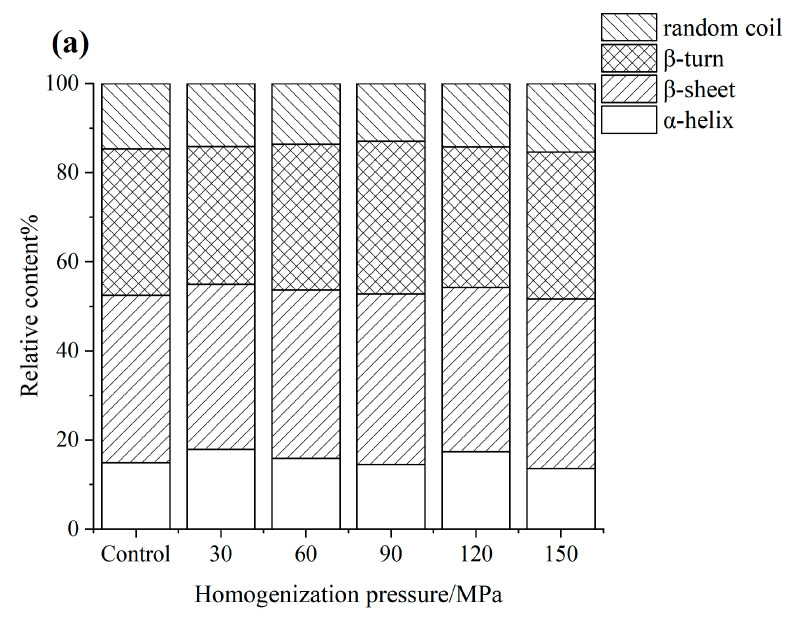
The effect of high pressure homogenization on the secondary structure of SPI (**a**), WPI (**b**), SPI-WPI (**c**) on the FTIR.

**Figure 3 foods-12-03358-f003:**
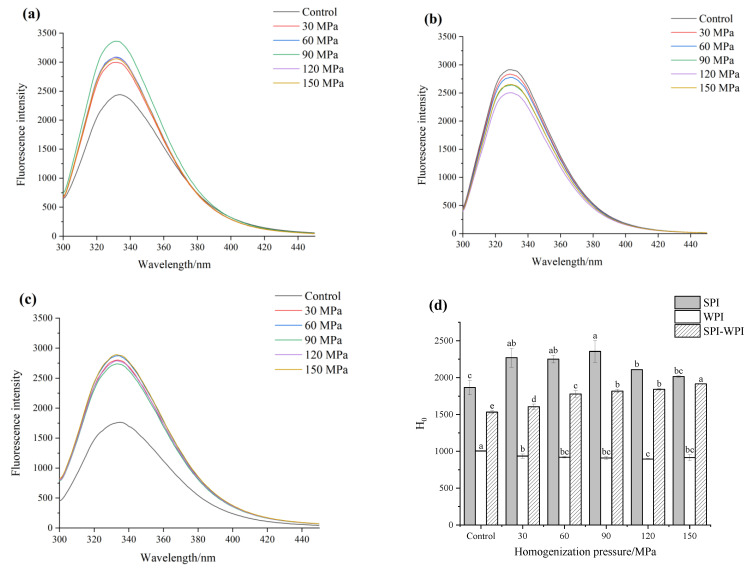
The effect of high pressure homogenization on the fluorescence spectra of SPI (**a**), WPI (**b**), SPI-WPI (**c**), and surface hydrophobicity (**d**) of SPI, WPI, SPI-WPI. Different lower case letters in Figure (**d**) indicate a significant difference (*p* < 0.05).

**Figure 4 foods-12-03358-f004:**
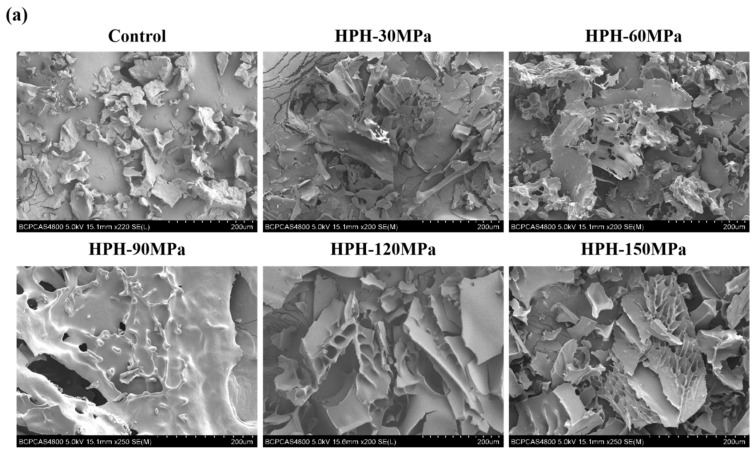
Microstructure of SPI (**a**), WPI (**b**) and SPI-WPI (**c**) under different homogenization pressures treatment.

**Figure 5 foods-12-03358-f005:**
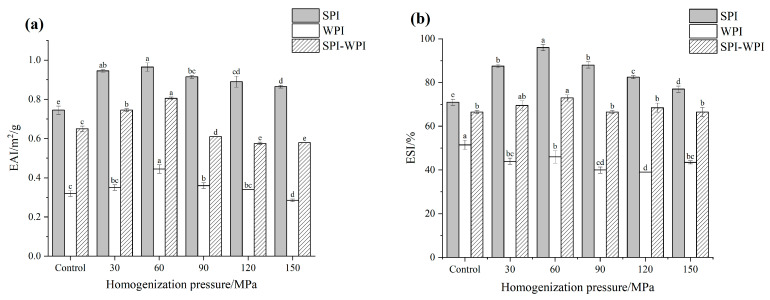
The effect of different homogenization pressures on the EAI (**a**), ESI (**b**), surface tension with time (**c**), surface pressure with time (**d**), curves of surface pressure with square roots of time (**e**), curves of proteins absorbed on the oil-water surface (**f**) of SPI, WPI, SPI-WPI. Different lower case letters in Figures (**a**,**b**) indicate a significant difference (*p* < 0.05).

**Figure 6 foods-12-03358-f006:**
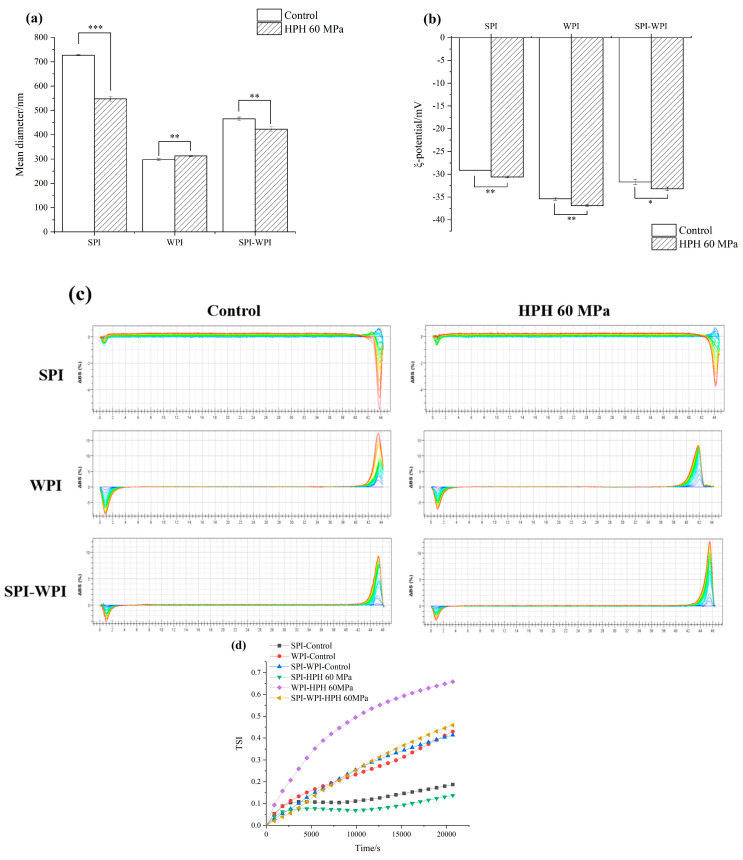
The effect of high pressure homogenization on the mean diameter (**a**), zeta potential (**b**), changes in backscattering profiles (**c**), Turbiscan stability index (**d**), the curves of MSD-time (**e**) of SPI, WPI, SPI-WPI emulsion containing 10% (*w*/*w*) oil. As the scanning time flows, the color of the line gradually changes from blue to green and eventually to red. Different lower case letters in Figures (**a**,**b**) indicate a significant difference (*p* < 0.05). “*, **, ***” indicate the significant difference, which “*” means “0.01 < *p* < 0.05”; “**” means “0.001 < *p* < 0.01”; and “***” means “*p* < 0.001”.

**Table 1 foods-12-03358-t001:** The effect of high pressure homogenization on the proteins absorbed on the oil-water surface.

	Control			HPH 60 MPa		
	SPI	WPI	SPI-WPI	SPI	WPI	SPI-WPI
K_diff_	5.89 × 10^−2^ ± 0.003 ^a^	7.83 × 10^−2^ ± 0.002 ^b^	6.57 × 10^−2^ ± 0.002 ^a^	5.71 × 10^−2^ ± 0.001 ^b^	8.67 × 10^−2^ ± 0.001 ^a^	6.30 × 10^−2^ ± 0.004 ^a^
K_p_	1.19 × 10^−3^ ± 0.001 ^a^	1.26 × 10^−3^ ± 0.000 ^b^	1.21 × 10^−3^ ± 0.002 ^a^	1.25 × 10^−3^ ± 0.001 ^a^	1.72 × 10^−3^ ± 0.001 ^a^	1.22 × 10^−3^ ± 0.002 ^a^
K_r_	1.23 × 10^−2^ ± 0.001 ^b^	1.28 × 10^−2^ ± 0.001 ^a^	2.30 × 10^−2^ ± 0.002 ^a^	1.75 × 10^−2^ ± 0.001 ^a^	1.08 × 10^−2^ ± 0.002 ^b^	9.90 × 10^−3^ ± 0.001 ^b^

Different lower case letters in Table 1 indicate a significant difference (*p* < 0.05).

**Table 2 foods-12-03358-t002:** The effect of high pressure homogenization on the proteins on the elasticity index (EI), macroscopic viscosity index (MVI) and solid-liquid balance (SLB) of SPI, WPI and SPI-WPI emulsion of 10% (*w*/*w*) oil phase concentration.

	Control			HPH 60 MPa		
	SPI	WPI	SPI-WPI	SPI	WPI	SPI-WPI
EI	6.99 × 10^−4^ ± 0.000 ^a^	4.60 × 10^−4^ ± 0.001 ^a^	4.73 × 10^−4^ ± 0.000 ^b^	6.86 × 10^−4^ ± 0.001 ^b^	3.74 × 10^−4^ ± 0.002 ^b^	5.09 × 10^−4^ ± 0.000 ^a^
MVI	1.26 × 10^−5^ ± 0.000 ^a^	1.61 × 10^−5^ ± 0.000 ^b^	1.54 × 10^−5^ ± 0.001 ^a^	4.22 × 10^−6^ ± 0.000 ^b^	2.06 × 10^−5^ ± 0.000 ^a^	4.47 × 10^−6^ ± 0.001 ^b^
SLB	7.87 × 10^−1^ ± 0.002 ^b^	5.16 × 10^−1^ ± 0.001 ^a^	6.00 × 10^−1^ ± 0.001 ^b^	1.02 ± 0.001 ^a^	3.58 × 10^−1^ ± 0.001 ^b^	7.30 × 10^−1^ ± 0.001 ^a^

Different lower case letters in Table 2 indicate a significant difference (*p* < 0.05).

## Data Availability

The datasets generated for this study are available on request to the corresponding author.

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
