# Peer review of "The Effect of High Pressure Homogenization on the Structure of Dual-Protein and Its Emulsion Functional Properties"

_foods, 2023, doi:10.3390/foods12183358_

Round 1

Reviewer 1 Report

The article submitted for review presents very good quality in almost every element. Below are my comments on specific sections of the article.

line 63- please make here another paragraph

In the introduction, the authors explained the principle of HPH and described the latest information on protein modification, but forgot to mention that originally HPH is used for very deep homogenization of the product and is one of the methods of preservation liquid products, most likely juices. Here are some examples: https://doi.org/10.1016/j.foodchem.2021.131023 ; https://doi.org/10.3390/molecules28052018.

The purpose and scope of the work are clearly defined.

2.2. Preparation of Protein Solution and HPH treatment - what was the input temperature and whether the samples were cooled or adjusted to the specified input temperature after each cycle? please state this in the text

Table 1 and 2 - statistical processing of values is missing

The authors have graphically presented the results of their study very nicely. They did not miss any determination in the discussion of the results. The discussion was conducted fairly and in depth, also with reference to the literature.

The authors wrote out the most important conclusions from the experiment. All conclusions resulted from the authors' work. They are a factual and informative account of the most important findings.

Author Response

Dear Editors and Reviewers:

We are pleased to have received your comments on this manuscript (Foods-2567196) entitled “The Effect of High Pressure Homogenization on the Structure of Dual-protein and its Emulsion Functional Properties”. Thank you very much for giving us an opportunity to revise our manuscript. According to these comments, we carefully checked the first version and finished all necessary correction in the revision. In addition, point-by-point responses to the comments are listed below this letter.

With these modifications, we believe the quality of this manuscript has been improved, and please consider it for publication in Foods. We thank the reviewers and the editor for their constructive comments.

If you have any question, please contact us without hesitation.

With best regards

Yours sincerely,

Jing Wang & Xiaoye He

E-mail: wangjing07@caas.cn & hexiaoye@caas.cn  

Response to Reviewer 1 Comments

Point 1: line 63- please make here another paragraph

In the introduction, the authors explained the principle of HPH and described the latest information on protein modification, but forgot to mention that originally HPH is used for very deep homogenization of the product and is one of the methods of preservation liquid products, most likely juices. Here are some examples: https://doi.org/10.1016/j.foodchem.2021.131023 ; https://doi.org/10.3390/molecules28052018.

Response 1: Thank you for your suggestion. We have read the references mentioned in detail which was very useful for us to refine the background of HPH. We have added the application of HPH in fruit and vegetable juice processing, and introduced the side effect which might cause temperature increase, to make the background of HPH more comprehensive (Line 61-63, 70-74 in revised manuscript).

Point 2: 2.2. Preparation of Protein Solution and HPH treatment-what was the input temperature and whether the samples were cooled or adjusted to the specified input temperature after each cycle? please state this in the text.

Response 2: Thank you for your suggestion. The input temperature of protein samples were about 25℃ (room temperature). The high-pressure homogenizer was connected with a condenser to keep the temperature of protein solutions at 18±1℃ after each cycle. These preparation methods were added in our manuscript (Line 99-105 in revised manuscript).

Point 3: The purpose and scope of the work are clearly defined.

Response 3: Thanks for the positive comments.

Point 4: Table 1 and 2 - statistical processing of values is missing.

Response 4: Thank you for your suggestion. The statistical processing of values were added in Table 1 and 2.

Point 5: The authors have graphically presented the results of their study very nicely. They did not miss any determination in the discussion of the results. The discussion was conducted fairly and in depth, also with reference to the literature.

Response 5: We would like to express our sincere appreciate to you for your positive comment to our results analysis!

Point 6: The authors wrote out the most important conclusions from the experiment. All conclusions resulted from the authors' work. They are a factual and informative account of the most important findings.

Response 6: We are very grateful for your praise once again, your encouragement will become the strong motivation for our research in the future!

Reviewer 2 Report

This manuscript describes the effect of high pressure homogenization (HPH) process on gels made of soy-protein isolate (SPI), whey-protein isolate (WPI) and their mixture (SPI-WPI).

The motivation and methodology seems in several ways to follow that of ref. 35 (who studied the effect of ultra-sonication on similar gels). As such, this may interest the food science and engineering community, albeit incrementally.

There are several issue that need to be clarified (in order of appearances in the manuscript, not of importance):

1.      In the abstract, please spell out SPI and WPI

2.      Line 53 claims a “non-thermal” process. The authors should show their verification that no heating occurred during HPH, which can induce significant heating.

3.      Line 89: what was the duration of each HPH cycle?

4.      Section 2.3.6 Surface Hydrophobicity.

a)      The authors provide ref. 20 for the analysis method. This is over 40 years old,and uses a different probe (cis-parinaric acid) and different procedure. In fact, the method used hear (with ANS) is quite similar to that used in ref. 35 (from 2020). Why not quote this as well?

b)     It would be good to show, at least as supplemental information, the graphs of intensity vs. protein concentration from which the hydrophobicity Index Ho is calculated.

c)      The “hydrophobicity” is termed “surface hydrophobicity” and labelled Ho in the plots (e.g. section 3.6 and Fig. 3), but in the methods section (2.3.6) it is termed “surface hydrophobicity Index” without any notation, and no equation is provided for its calculation. By the way, this term is labelled So in references 20 and 35 (it’s OK to change symbols…)

d)     Line 131: please spell out ANS (in its first appearance)

5.      Section 2.3.7: please provide type of detection used ( I assume secondary electrons, detector type?)

6.      Lines 155, 156: equations should be numbered, and referred to by number in the subsequent text (?)

7.      Line 153: Can the authors comment on why this pressure (50 MPA) was used for emulsion preparation?

8.      Line 201: It is not clear if the authors mean 24 h after treatment or after 24 h of treatment (I doubt HPH was employed for 24 h).

9.      Fig. 1a: What are the different bottles? The labels are illegible.

10.   ACCURACY: Throughout the text the authors claim results (e.g. solubility, IR intensity, hydrophobicity) with sometimes four and five significant figures. Unless the authors can validate such accuracy, this should be amended. Some kind of error analysis would be useful.

11.   In particular, Line 286: what is the significance of the statement: “the random coil decreased from 14.73% to 14.19%”. Is this difference significant? What is significant in this study?

12.   Line 312: what is “hair groups” ?

13.   Section 3.7:

a)      Fig. 4: The images are too small to observe the structures that are reported in the text. Also the scale bars are not legible.

b)      Lines 368-369 claim that ref 35 observed similar changes in structure. The relevant images in ref 35 are quite different, being much more homogeneous in appearance, not showing any lamellar structure. So what is the similarity? Also the English usage in this sentence is not clear.

14.   Section 3.8.2: The title states surface tension, the measurement of which is provided in section 2.4.2. The text in 3.8.2 and Fig. 5 d,e,f refer also to “surface pressure, and the text to “interfacial pressure”. Also, some logarithmic function of “surface pressures” is presented in Fig. 5f. It is not clear what these are, how calculated and what is their meaning.

15.   Line 423: Why does plot of “surface pressure” vs. time^1/2 correspond to “diffusion constant” (and what units??)

Author Response

Dear Editors and Reviewers:

We are pleased to have received your comments on this manuscript (Foods-2567196) entitled “The Effect of High Pressure Homogenization on the Structure of Dual-protein and its Emulsion Functional Properties”. Thank you very much for giving us an opportunity to revise our manuscript. According to these comments, we carefully checked the first version and finished all necessary correction in the revision. In addition, point-by-point responses to the comments are listed below this letter.

With these modifications, we believe the quality of this manuscript has been improved, and please consider it for publication in Foods. We thank the reviewers and the editor for their constructive comments.

If you have any question, please contact us without hesitation.

With best regards

Yours sincerely,

Jing Wang & Xiaoye He

E-mail: wangjing07@caas.cn & hexiaoye@caas.cn

Response to Reviewer 2 Comments

Point 1: This manuscript describes the effect of high pressure homogenization (HPH) process on gels made of soy-protein isolate (SPI), whey-protein isolate (WPI) and their mixture (SPI-WPI).

The motivation and methodology seems in several ways to follow that of ref. 35 (who studied the effect of ultra-sonication on similar gels). As such, this may interest the food science and engineering community, albeit incrementally.

Response 1: Thanks for the positive comments.

Point 2: In the abstract, please spell out SPI and WPI.

Response 2: Thank you for your suggestion. The full name of SPI and WPI have been spelled out in abstract (Line 16-17 in revised manuscript).

Point 3: Line 53 claims a “non-thermal” process. The authors should show their verification that no heating occurred during HPH, which can induce significant heating.

Response 3: Thanks for your suggestion. Generally, the product temperature rises by 2.0-3.2℃ for every 10 MPa of pressure using HPH, therefore, a condenser was used  to keep the temperature of the samples at 18±1℃ (Line 72-74 and 102-105 in revised manuscript).

Point 4: What was the duration of each HPH cycle?

Response 4: The duration of each HPH cycle is about 13 s, which has been added in the revised manuscript (Line 102 in revised manuscript).

Point 5(a): The authors provide ref. 20 for the analysis method. This is over 40 years old, and uses a different probe (cis-parinaric acid) and different procedure. In fact, the method used here (with ANS) is quite similar to that used in ref. 35 (from 2020). Why not quote this as well?

Response 5(a): Thanks for your suggestion. Revised accordingly (Line 114-115 and 650-652 in revised manuscript).

Point 5(b): It would be good to show, at least as supplemental information, the graphs of intensity vs. protein concentration from which the hydrophobicity Index H0 is calculated.

Response 5(b): Thanks for your suggestion. The graphs of fluorescence intensity vs. protein concentration have been added in supplementary materials (Fig. S1), and the slope of each curve was the H0 of protein samples.

Point 5(c): The “hydrophobicity” is termed “surface hydrophobicity”and labelled H0 in the plots (e.g. section 3.6 and Fig. 3), but in the methods section (2.3.6) it is termed“surface hydrophobicity index”without any notation, and no equation is provided for its calculation. By the way, this term is labelled S0 in references 20 and 35 (it’s OK to change symbols…)

Response 5(c): Thanks for your suggestion. I'm sorry for your confusion caused by my inappropriate writing. The surface hydrophobicity index (H0) is the slope of the graphs of intensity vs. protein concentration (Line 148-149, 152 and 360-369 in revised manuscript), and the H0 is generally used to reflect the surface hydrophobicity of the samples.

Point 5(d): Please spell out ANS (in its first appearance).

Response 5(d): Thanks for your suggestion. Revised accordingly (Line 144-145 in revised manuscript).

Point 6: Section 2.3.7: please provide type of detection used (I assume secondary electrons, detector type?)

Response 6: Thanks for your suggestion. Revised accordingly (Line 158-159 in revised manuscript).

Point 7: Lines 155, 156: equations should be numbered, and referred to by number in the subsequent text (?)

Response 7: Thanks for your suggestion. Revised accordingly (Line 172, 397, 432 and 444-445, in revised manuscript).

Point 8: Line 153: Can the authors comment on why this pressure (50 MPA) was used for emulsion preparation?

Response 8: According to preliminary experiment and some references such as Hogan et al.(2001) [DOI:10.1021/jf000276q], Yan et al.(2017) [DOI:10.1111/jfpe.12578] and Chia et al.(2019) [DOI: 10.1016/j.foodres.2019.02.026], the pressure 50 MPa was used for emulsion preparation.

Point 9: Line 201: It is not clear if the authors mean 24 h after treatment or after 24 h of treatment (I doubt HPH was employed for 24 h).

Response 9: Thanks for your suggestion. I'm sorry for your confusion caused by my inappropriate writing. After HPH treatment, the appearance changes of the protein solutions were observed after standing for 24 h. Revised accordingly (Line 217-218 in revised manuscript).

Point 10: Fig. 1a: What are the different bottles? The labels are illegible.

Response 10: I’m sorry for giving illegible labels in Fig.1. We have added the label above each bottle to make the information in the figure more specific and intuitive.

Point 11: ACCURACY: Throughout the text the authors claim results (e.g. solubility, IR intensity, hydrophobicity) with sometimes four and five significant figures. Unless the authors can validate such accuracy, this should be amended. Some kind of error analysis would be useful.

Response 11: Thanks for your suggestion. All significant figures in the data (e.g. solubility, IR intensity, hydrophobicity) have been rounded to two decimal places. (Line 260-261, 268-269, 272, 301-305, 315-316, 319-320, and 357-366 in revised manuscript).

Point 12: In particular, Line 286: what is the significance of the statement: “the random coil decreased from 14.73% to 14.19%”. Is this difference significant? What is significant in this study?

Response 12: Thanks for your suggestion. It is found that as the pressure increased to 30 MPa, the relative content of random coil in SPI had no significant decrease tendency (from 14.73% to 14.19%), whereas the relative content of α-helix increased from 14.91% to 17.86%, which was sufficient to indicate that the hydrogen bond interaction enhanced at this time. It has been revised to “The majority of secondary structures of SPI were β-sheet and β-turn, and the relative content of α-helix increased from 14.91% to 17.86% under 30 MPa treatment, indicating that the hydrogen bond interaction enhanced, which informed the highly ordered protein structure [34].” (Line 300-303 in revised manuscript).

Point 13: What is“hair groups”?

Response 13: It has been revised to “chormophoric groups” (Line 328 in revised manuscript).

Point 14(a): Fig.4: The images are too small to observe the structures that are reported in the text. Also the scale bars are not legible.

Response 14(a): The microstructure images and scale bars under different pressure treatment have been magnified (Fig. 4).

Point 14(b): Lines 368-369 claim that ref 35 observed similar changes in structure. The relevant images in ref 35 are quite different, being much more homogeneous in appearance, not showing any lamellar structure. So what is the similarity? Also the English usage in this sentence is not clear.

Response 14(b): Thanks for your suggestion. I agree with your comments. Revised accordingly (Line 373-380 in revised manuscript).

Point 15: Section 3.8.2: The title states surface tension, the measurement of which is provided in section 2.4.2. The text in 3.8.2 and Fig. 5 d,e,f refer also to “surface pressure, and the text to “interfacial pressure”. Also, some logarithmic function of “surface pressures” is presented in Fig. 5f. It is not clear what these are, how calculated and what is their meaning.

Response 15: Thanks for your suggestion. I'm sorry for your confusion caused by my inappropriate writing. Surface pressure is mainly used to reflect the additional pressure of emulsifier in the adsorption process of oil-water interface. We have revised the description as “surface pressure” uniformly in 3.8.2 text (Line 405-414 and 446 in revised manuscript) and Fig.5. The equation to calculate the relevant parameter of interface adsorption kinetics like Kdiff, Kp and Kr values have been added (Line 431-452 in revised manuscript).

Point 16: Line 423: Why does plot of “surface pressure” vs. time^1/2 correspond to “diffusion constant” (and what units??)

Response 16: Thank you for your question. According to Li et al.(2022), in the process of protein adsorption at the oil-water interface, the initial state is that protein transfer from water phase to the oil phase through diffusion. Generally, using the equation modified by Ward and Tordai (1946) could calculate the protein diffusion phase:

If the diffusion of the emulsifier at the interface dominates the adsorption kinetics, the Π-t1/2 should be linear, and the slope is denoted by Kdiff. We have added the description in our manuscript to explain the interface adsorption kinetics of protein under HPH treatment (Line 437-440 in revised manuscript).
